# Increased Fat Taste Preference in Progranulin-Deficient Mice

**DOI:** 10.3390/nu13114125

**Published:** 2021-11-17

**Authors:** Lana Schumann, Annett Wilken-Schmitz, Sandra Trautmann, Alexandra Vogel, Yannick Schreiber, Lisa Hahnefeld, Robert Gurke, Gerd Geisslinger, Irmgard Tegeder

**Affiliations:** 1Institute of Clinical Pharmacology, Faculty of Medicine, Goethe-University Frankfurt, 60590 Frankfurt, Germany; schumann@med.uni-frankfurt.de (L.S.); wilken-schmitz@em.uni-frankfurt.de (A.W.-S.); trautmann@med.uni-frankfurt.de (S.T.); a.vogel@med.uni-frankfurt.de (A.V.); Yannick.Schreiber@itmp.fraunhofer.de (Y.S.); hahnefeld@med.uni-frankfurt.de (L.H.); robert.gurke@itmp.fraunhofer.de (R.G.); geisslinger@em.uni-frankfurt.de (G.G.); 2Fraunhofer Institute for Translational Medicine and Pharmacology ITMP, 60596 Frankfurt, Germany; 3Fraunhofer Cluster of Excellence for Immune Mediated Diseases (CIMD), 60596 Frankfurt, Germany

**Keywords:** taste buds, lipids, progranulin, CD36, IntelliCage

## Abstract

Progranulin deficiency in mice is associated with deregulations of the scavenger receptor signaling of CD36/SCARB3 in immune disease models, and CD36 is a dominant receptor in taste bud cells in the tongue and contributes to the sensation of dietary fats. Progranulin-deficient mice (Grn^−/−^) are moderately overweight during middle age. We therefore asked if there was a connection between progranulin/CD36 in the tongue and fat taste preferences. By using unbiased behavioral analyses in IntelliCages and Phenomaster cages we showed that progranulin-deficient mice (Grn^−/−^) developed a strong preference of fat taste in the form of 2% milk over 0.3% milk, and for diluted MCTs versus tap water. The fat preference in the 7d-IntelliCage observation period caused an increase of 10% in the body weight of Grn^−/−^ mice, which did not occur in the wildtype controls. CD36 expression in taste buds was reduced in Grn^−/−^ mice at RNA and histology levels. There were no differences in the plasma or tongue lipids of various classes including sphingolipids, ceramides and endocannabinoids. The data suggest that progranulin deficiency leads to a lower expression of CD36 in the tongue resulting in a stronger urge for fatty taste and fatty nutrition.

## 1. Introduction

Progranulin is a multifaceted, neurotrophic and anti-inflammatory protein, which is essential for neuronal survival and regeneration after axonal damage [1,2]. Loss of progranulin in mice leads to synaptic pruning by activated microglia [3]. In humans, dominant negative loss-of-function mutations are associated with frontotemporal dementia (FTD) [4,5], whereas homozygous loss-of-function mutations cause neuronal ceroid lipofuscinosis [6]. In the periphery, progranulin reduces inflammation [7,8] by increasing the phagocytosis activity of myeloid immune cells [9,10] and possibly via anti-TNF signalling [11,12]. However, progranulin promotes the antigen-stimulated activation of T-cells in the context of autoimmunity [10,13,14]. Corresponding to the duality, progranulin reduces the inflammation of obese adipose tissue in some studies [15,16], but increases insulin resistance, interleukin-6 and inflammation in obese fat tissue in other studies [17,18,19]; high serum levels of progranulin are associated with obesity [20,21]. Similar to other adipokines, progranulin promotes insulin resistance in experimental models [17,22,23,24]. However, its loss rather than progranulin overexpression was associated with unfavourable metabolic features in mice such as body weight gain, low voluntary exercise [25], increased atherosclerosis and cholesterol accumulation in macrophages [26].

Progranulin-deficient mice are moderately overweight and on a standard lab diet throughout middle age [25], but the behavioural manifestations of frontotemporal dementia such as overactivity, impulsivity, over-consumption of sweet drinks, loss of sociability and learning difficulties occur later [25,27,28], suggesting that being overweight is not easily attributable to compulsive feeding in the context of the FTD phenotype. It was shown that progranulin-deficient myeloid immune cells adopt a Gaucher-cell-like phenotype upon stimulation [29]. These cells are lipid-laden myeloid immune cells which accumulate glucosylceramides and sphingolipids in the lysosome owing to an enzymatic defect of glucocerebrosidase (GBA1) [30,31,32]. Progranulin-deficient immune cells do not share the enzymatic defect but are required for GBA1 functioning [33] and likewise accumulate lipids in the lysosome suggesting alterations of lipid degradation. Indeed, we have recently observed that the expression of the scavenger class B type receptor (SCARB3), alias cluster of differentiation 36 (CD36), depended on progranulin, suggesting a functional link between progranulin and CD36 [10,34].

CD36 is also known as fatty acid translocase. It is a transmembrane receptor for oxidized and non-oxidized lipoproteins and long-chain fatty acids. Upon ligand binding, CD36 is internalized along with the ligand, which is then transferred to the lysosome. It is well accepted that progranulin is required for the efficient autophagolysosomal degradation of lipids and other cargo [1,35,36,37] suggesting that it may interfere with the paths of CD36. Low-function or low-expression polymorphisms of CD36 were associated with obesity [38,39] owing to a reduced lipid uptake by scavenger cells and a loss of the oro-sensory perception of fatty nutrients, i.e., fat taste [40,41,42,43,44,45]. CD36 is highly expressed in taste bud cells in fish [46], rodents [47] and humans [48,49,50] and is required for the sensing of fatty acids in food [40,51]. It was observed that a high-fat diet or obesity resulted in a loss of CD36 expression in taste bud cells [47], and its expression was low in models of obesity such as ghrelin-deficient mice [52]. Fat sensing is contributed by the G-protein-coupled receptor, GPR120 [42,53,54], calcium sensing via stromal interaction molecule (STIM1) and calcium influx through transient receptor potential, TRPC3 ion channels [55]. In addition, the signalling through cannabinoid CB1 receptors of taste bud cells contributes to the gustatory sensing of sweet [56,57,58] and fat [59], and contributes to the hedonic nature of pleasurable food [60,61]. Hence, endocannabinoids and further lipid species likely contribute to the complex oro-sensory detection and pleasure of fat in food, which may be modulated by progranulin.

Considering the mild overweight and previously observed deregulation of CD36 in progranulin-deficient mice, we hypothesized that progranulin was a regulator of fat taste via CD36. To address the hypothesis, we studied fat taste preferences of progranulin-deficient mice in two-choice behavioural paradigms using IntelliCage and Phenomaster observations and assessed the expression of CD36 and of bioactive lipids in taste buds.

## 2. Materials and Methods

### 2.1. Progranulin-Deficient and Control Mice

Progranulin knockout mice (Grn^−/−^) [62] were maintained as homozygous colony. Sex- and age-matched Grn-flfl mice were used as controls, as described [1,2]; they had identical C57BL6 genetic backgrounds. Mice were raised and maintained in the same breeding facility and kept in the same room for life. They had free access to food and water and were maintained in climate-controlled rooms at a 12 h light-dark cycle. The ages, sex and numbers are given in the respective figure legends. The experiments followed the “Principles of laboratory animal care” (NIH publication No. 86-23, revised 1985). They were approved by the local Ethics Committee for animal research (Regierungspräsidium Darmstadt, Germany; V-54-19 c 20/12-FK1074, approval date 10 October 2016), agreed with the 3R ARRIVE guideline and were in line with the European and German regulations for animal research (GV-SOLAS).

### 2.2. IntelliCage Description and Setup

The IntelliCage (NewBehavior AG, Zurich, Switzerland) [25,63,64,65,66] consisted of four operant corners, each with two water bottles, sensors, light-emitting-diodes (LEDs) and doors that controlled the access to the water bottles. The system fit into a large cage (20 cm × 55 cm × 38 cm, Tecniplast, Germany, 2000P) and housed 16 mice per cage. Triangular red shelters served as sleeping quarters and as stands to reach the food. The floor was covered with standard bedding. Mice were tagged with radio-frequency identification (RFID)-transponders, which were read with an RFID antenna that was integrated in the corner entrance. Inside the corners were two doors that could provide access to water bottles. Mice had to make nosepokes (NP) to open the doors for water access. The IntelliCage was controlled by a computer with IntelliCage Plus software (TSE, Berlin, Germany), which executed pre-programmed experimental tasks and schedules and allowed for live observation and data visualization. The numbers and duration of corner visits, nosepokes, and licks were recorded. Data analysis was completed with FlowR (XBehavior, Zürich, Switzerland).

### 2.3. IntelliCage Tasks and Schedules and Analysis of Fat Taste Preference

The first set of experiments employed a two-choice paradigm in IntelliCages, which allowed for an unbiased observation of the behaviour in groups of mice without handling. In each of the four operant corners, mice could freely choose between two drinking bottles with either 0.3% or 2% milk. The 0.3% side was defined as “side error”. The free-choice module was active for 2 × 3 h daily. In between (default module) doors remained closed. The milky session was interrupted by two days tap water.

Female mice (*n* = 15–16 per genotype) were adapted to the IntelliCage before starting the fat taste preference module, which was referred to as “FatLiking”. During adaptations mice learnt to make a nosepoke to open a door for 5 s to gain access to the drinking bottle. The nosepoke protocol was turned on from 11–14:00 h and from 2–5 h each day. During these times, a nosepoke opened the door. Outside of these times, the doors remained closed, i.e., the nosepoke protocol was OFF and the default module was ON. The day patterns were maintained during the fat preference testing period. Drinking bottles were filled with milk instead of tap water. One side in each corner was 2% milk (fatty), the other 0.3% skimmed milk. Mice could freely choose the side, and NP opened the door for 5 s on this side. NP on the other side had no effect. To drink more, mice had to leave the corner and start a new visit. After three days, milk was switched to tap water for two days, and then back to milk for another two days. The total observation comprised seven days. The fat-milk side was defined as correct. Hence, a side error was a nosepoke on a door giving access to skimmed milk. Body weights were obtained before and after the observation period.

### 2.4. Phenomaster Analysis of Fat Taste Preference

Five Grn^−/−^ and five Grn-flfl mice were used in the Phenomaster experiment. The mice were 9–10 weeks old. The Phenomaster (TSE, Bad Homburg, Germany) offers an automated, metabolic and behavioural monitoring in home cage environments. Drinking and feeding behaviour and voluntary wheel running (VWR) were monitored. Mice were adapted to the drinking bottles for 5 days in their home cage. In the Phenomaster cage, mice had free access to water and standard diet pellets and free access to a voluntary running wheel. In the 1st Phenomaster test lasting 24 h, the drinking bottle was filled with tap water. In the second 24 h period, mice had access to 5% medium chain triglycerides (MCT) diluted in water (1:4 diluted 20% Lipofundin, B. Braun Melsungen AG, Germany). Drinking volumes and feeding were monitored with Phenomaster software-controlled precision scales of the drinking bottle and the food basket.

### 2.5. Immunofluorescence Analysis of CD36 in Taste Buds

After overnight fasting, Grn^−/−^ and Grn-flfl mice (*n* = 8–9, female and male) were terminally anaesthetized with carbon dioxide and perfused transcardially with cold 0.9% sodium chloride (NaCl) followed by 2.25% buffered paraformaldehyde (PFA) in saline. Tongues were removed, fixed in 2.25% PFA in 1× PBS and subsequently transferred into 20% sucrose. Tongues were cut to obtain 10 µm thick sections which were stored at −80 °C. Sections were mounted on glass slides. After three washes with 1× PBS, sections were permeabilized with PBST (1× PBS + 0.1% Triton-X-100). Blocking was performed at room temperature (RT) for 45 min in 3% bovine serum albumin (BSA) in PBST. Primary antibodies (Appendix A) were applied in PBST with 1% BSA at 4 °C overnight. After washing, specific binding was revealed with species-specific, fluorochrome-conjugated secondary antibodies, applied at RT for 2 h under light protection. Sections were washed in PBST and the next primary/secondary antibody staining was performed. Finally, sections were washed in 1× PBS and counterstained using the DNA marker DAPI, applied at a concentration of 10 µg/mL at RT for 10 min. After washing, sections were mounted in Aqua Poly Mount. Images were captured using an inverted fluorescence microscope (Carl Zeiss, ZEN software, Germany).

For quantification of immunofluorescence signals, images were submitted to the Particle Counter implemented in FIJI ImageJ after background subtraction and threshold setting using Li’s algorithm. At least 3 images were analyzed per mouse of *n* = 8–9 mice per group.

### 2.6. RNA Extraction and QRT-PCR Analysis of Fat Taste Receptors in the Tongue

Grn^−/−^ and Grn-flfl female and male mice (*n* = 9 per genotype) were sacrificed with carbon dioxide and debleeding, and tongues were removed. The tongue tip, lateral border and region of the circumvallate papillae (CVP) were separated, and the tissue was stored at −80 °C. Total RNA was extracted with Qiagen RNeasy spin column kit and quantified on a NanoDrop^®^ using A260/A280 and A260/A230 ratios. RNA was reverse transcribed with Thermo Scientific Verso first-strand cDNA synthesis kit using random hexamers as primers. The QRT-PCR was performed on a TaqMan instrument (Thermo Fisher Scientific, Germany, 384 block) using the SYBR Green technique. The primer sequences were CD36 FW 5′-TAG TAG GCG TGG GTC TGA AG-3′, CD36 RV 5′-GCT TCA GGG AGA CTG TTG AA-3′ GPR120 FW 5′-GTGACTTTGAACTTCCTGGTGCC-3′, GPR120 RV 5′-CAGAGTATGCCAAGCTCAGCGT-3′. PCR reaction and amplicon detection were performed with the QuantStudio TM 5 Real-Time PCR Design and Analysis Software v1.5.1 (Thermo Fisher Scientific). Cycle numbers of candidate genes were normalized to the cycle numbers of the housekeeping gene eukaryotic elongation factor 1 gamma-like protein (EEF1G), and quantification was conducted according to the relative 2^−∆∆Ct^ method versus control mice.

### 2.7. Analysis of Lipid Signalling Molecules in Tongue Tissue

Endocannabinoids were analyzed in plasma and tongue tissue using liquid chromatography-electrospray ionization-tandem mass spectrometry (LC-ESI-MS/MS), updated according to procedures described in [67,68,69,70,71]. Briefly, plasma samples as well as tissue homogenates (two different sample quantities equaling 2 and 0.05 mg of tissue) were extracted by liquid–liquid extraction using 400 µL of ethylacetate: n-hexane (9:1, *v*/*v*) after spiking them with the corresponding internal standards (deuterated analytes). The analysis was performed using a hybrid triple quadrupole-ion trap mass spectrometer QTRAP 6500+ (Sciex, Darmstadt, Germany) equipped with a Turbo-V-source operating in positive ESI mode. The mass spectrometer was coupled to an Agilent 1290 Infinity II UHPLC system equipped with an Acquity UPLC BEH C18 UPLC column (100 × 2.1 mm, 1.7 μm, Waters, Eschborn, Germany).

Sphingoid bases and ceramides were analyzed in plasma and in tongue tissue. Lipid analyses were conducted using LC-ESI-MS/MS, according to procedures described in detail in [71]. In brief, tissue samples were homogenized in ethanol:water (1:3, *v*/*v*) with a ball mill mixer and 10 zirconium oxide grinding balls (6500 rpm for 2 × 45 s at 10 °C). Subsequently, homogenates equaling 1 mg of tissue were extracted using a liquid–liquid extraction method. Plasma sample volumes were 10 μL. The analysis of all analytes was performed using a hybrid triple quadrupole-ion trap mass spectrometer QTRAP 5500 (Sciex, Darmstadt, Germany) equipped with a Turbo-V-source operating in positive ESI mode. The ceramides were separated using an Agilent 1290 Infinity I UHPLC system equipped with a Zorbax C18 Eclipse Plus UHPLC column (50 × 2.1 mm, 1.8 μm, Agilent technologies, Waldbronn, Germany). For separation of sphingoid bases a Zorbax Eclipse Plus C8 column (30 × 2.1 mm ID, 1.8 μm, Agilent technologies, Waldbronn, Germany) was used.

The analysis of endocannabinoids, as well as sphingoid bases and ceramides, was conducted in Multiple Reaction Monitoring (MRM) mode. Quality control samples of three different concentration levels (low, middle, high) were run as initial and final samples of each run after measuring a calibration curve. For all analytes, the concentrations of the calibration standards, quality controls and samples were evaluated by Analyst software 1.7.1 and MultiQuant Software 3.0.3 (both Sciex) using the internal standard method. Calibration curves were calculated by linear or quadratic regression with 1/x weighting or 1/x^2^ weighting. Variations in accuracy of the calibration standards were less than 15% over the range of calibration, except for the lower limit of quantification (LLOQ), where a variation in accuracy of 20% was accepted. Lipid concentrations are expressed as pg/mg of tissue or ng/mL plasma.

### 2.8. Statistical Analyses

Group data are presented as mean ± SD or 95% confidence interval (CI), the latter for behavioral time courses, specified in the respective figure legends. Data were analyzed with SPSS 25 and Graphpad Prism 9.2, Origin Pro 2021, and FlowR for IntelliCage experiments. Data were normally distributed according to the Shapiro–Wilk test. Groups were compared with unpaired, two-sided Student’s *t*-tests. Time course data or multifactorial data were submitted to two-way analysis of variance (ANOVA) using the factors ‘time’ and ‘group’. In case of significant differences, groups were mutually compared at individual time points using post hoc *t*-tests according to Dunnett, i.e., versus the control group, or according to Šidák. Post hoc comparisons for between-subject factors were not adjusted if they were predefined by the experiment (i.e., two genotypes). Asterisks in the figures show multiplicity-adjusted *p*-values.

In the IntelliCage, the number of visits and nosepokes were analyzed to assess overall activity, and the licks to assess drinking behavior and the success rate. Discriminant Principal Component Analysis (PCA) was used to reduce the complexity of behavioral parameters and assess the discrimination of the groups according to PCA scores. To assess the number of trials needed to achieve learning success, a probability test was used with the success criterion set to 0.35 for experiments with a random success of 0.25. Type 1 and type 2 errors were set to 0.05.

## 3. Results

### 3.1. Fat Milk Taste Preference of Progranulin-Deficient Mice in the IntelliCage

A two-choice paradigm was tested in IntelliCages to assess fat preferences. Mice could freely choose between two drinking bottles with either 0.3% or 2% milk, and skimmed milk was defined as an error. The free-choice module required nosepoking at the respective door and operated for 2 × 3 h daily. Outside of these times, the doors remained closed. The milky session was interrupted by two days tap water.

Progranulin-deficient mice were hyperactive during the milk sessions (Figure 1, Appendix A). They visited corners more frequently (Figure 1A, Appendix A) and made more licks (Figure 1C) on the 2% milk side, as indicated by a reduced proportion of “side errors” (Figure 1B, Appendix A). Hence, they showed a very clear preference of 2% milk over 0.3% milk and they gained about 10% of body weight (Figure 1F); however, the controls had no such preference and maintained their body weights. The overactivity of Grn^−/−^ mice returned to a normal behaviour during the tap water period, but was reinstated after switching back to milk (Figure 1A,C, Appendix A). The differences between genotypes were evident during the daytime and night time active module sessions. Outside of the times with active nosepoke protocol (i.e., doors remained closed), the frequency of visits and licks did not differ between Grn^−/−^ and control mice (Figure 1D,E; Appendix A). Throughout the observation, Grn^−/−^ mice made fewer nosepokes per visits (NP/visit ratio, Appendix A), which was a characteristic trait of Grn^−/−^ mice [25] and showed unfocused “running in-and-out” behaviour as early symptom of dementia. The clear preference of fatty milk and low errors was even more remarkable. The analysis of trials versus successes defined as nosepoke on the 2% milk door (Figure 1G,H) shows the very high frequency of trials of Grn^−/−^ mice leading to a doubling of successes as compared with the controls. Principal component analysis and clustering on the basis of behavioural readouts allowed for a clear separation of genotypes (Appendix A) and prediction of group membership.

### 3.2. MCT Fat Taste Preference of Progranulin-Deficient Mice in Phenomaster Cages

Since the preference of 2% milk over 0.3% milk might be influenced by sweetness, we additionally assessed the fat preference using a solution of medium-chain triglycerides (MCT) used for intravenous feeding in humans (Figure 2), which was “tasteless” and not sweet. We used Phenomaster cages, which allowed for a quantification of fluid intake. Drinking and feeding were identical in both genotypes when tap water was provided, but Grn^−/−^ mice drank significantly more when the drinking fluid was switched to 5% MCT, again showing the preference of fat taste. Food intake dropped during the MCT period in both genotypes showing that both genotypes were sensing the calories. Voluntary wheel running also dropped but did not significantly differ owing to a high interindividual variability.

### 3.3. Reduced Expression of CD36 in Taste Buds of Progranulin-Deficient Mice

Fat taste is supposed to be mediated through CD36, which is highly expressed in gustatory cells of taste buds, particularly in circumvallate papillae, and we have previously shown that progranulin regulates CD36 expression in immune cells [10]. Consequently, we studied the expression of CD36 in the tongues of Grn^−/−^ versus Grn-flfl mice. Immunofluorescence analyses of CVP revealed a high expression of CD36 in control mice, as expected. CD36 immunofluorescence (IFL) was significantly reduced in corresponding the regions of Grn^−/−^ mice (Figure 3A–C; Appendix A). Comparisons of the relative area covered with CD36 IFL (Figure 3C) revealed significantly lower CD36 in Grn^−/−^ tongues (Figure 3C), and a lower CD36 versus gustducin (GNAT3) ratio, which is a marker for taste bud cells. Correspondingly, a QRT-PCR analysis showed lower CD36 mRNA expression in CVP and tongue tips of Grn^−/−^ mice, whereas expression in lateral borders was alike in both genotypes (Figure 4A). The QRT-PCR results were confirmed in a second set of mice (Figure 4B), where only CVP and tongue tips were analyzed to increase analytical accuracy. In contrast to CD36, we did not observe differences in the GRP120 mRNA levels (Figure 4C) and cycle numbers for GPR120 were high (Ct range 36.4–39.7), indicating a low expression of GPR120, which is another fat taste receptor, differently regulated by dietary fats. Appendix A shows the ∆Ct values of CD36 and GPR120 versus the housekeeping gene EEF1G to reveal the expression levels.

CD36 expression analyses suggested that Grn^−/−^ mice may have a lower sensation of fat nutrients due to low CD36, and thus a stronger urge to take up fatty fluids. The taste of fatty nutrients may be influenced by lipids in saliva and the surrounding tissue. In particular, signaling of endocannabinoids via cannabinoid CB1 receptors affects sweet and fatty taste. We studied the concentrations of lipid candidates including endocannabinoids (Figure 5A), sphingoid base lipids (Figure 5B) and ceramides (Figure 5C) of different chain lengths and saturations in plasma and tongue tissue samples. Lipid patterns were identical in both genotypes in plasma (Figure 5 left panels) and tongue (Figure 5 right panels). The similarity of plasma lipid species shows that Grn^−/−^ mice have no major metabolic difference. We conclude that the urge for fat is caused at least in part via alterations of fat taste owing to a lower CD36 expression.

## 4. Discussion

We show in the present study that progranulin-deficient mice prefer 2% fat milk over skimmed milk, whereas control mice show no preference and also do not prefer milk over tap water. We have chosen milk rather than a solution of oleic acid or other single fatty acids because milk is a natural composition of flavours and a natural nutrient relevant for human nutrition. Although a lactose and protein composition is similar (4.8–5 g sugar; 3–3.3 g protein), 2% milk and 0.3% milk may differ in physicochemical properties. We therefore confirmed the fat preference with an MCT solution which was “tasteless” to the human taste buds. Again, progranulin-deficient mice preferred the fat-containing solution, whereas feeding between the groups of mice remained similar. The behavioural studies were conducted with unbiased PC-controlled, home-cage-based observation systems that were free of observer bias and allowed for observations of groups of mice. To further strengthen the results, a milk–water–milk schedule was used. This allowed for the observation of preference development, loss of preference during tap water and re-instatement of milk preference in the final period. The experiment was performed in approximately 6-month-old mice. At this age, the body weights of progranulin-deficient mice are still in the normal range and do not show abnormal behaviour suggestive of a development of frontotemporal dementia, such as overactivity, compulsiveness, or learning difficulties [25,27,28,63]. However, a high ratio of nosepokes per visit suggested a subtle early manifestation of dementia-like behaviour. Low NP/visit ratios indicate lower goal-directed explorations of the doors from the mice after corner entry [25]. It is even more remarkable that the mice developed a clear preference for 2% fat milk.

Because our previous studies suggested that progranulin regulated CD36 expression [10] we focused mechanistically on this fat taste receptor. Its expression was reduced in CVP and the anterior tongues of progranulin-deficient mice, whereas mRNA expression in the lateral tongue was similar. The expression pattern agreed with the predominant localization of CD36 in CVP in mice [72]. The observed association of low CD36 and high fat liking in progranulin-deficient mice was in accordance with previous studies, which revealed that CD36 expression in taste buds decreased upon high-fat dieting [47] and was permanently low in models of obesity, such as ghrelin knockout [52]. The activity of lipases and release of fatty acids from complex triglycerides in food leads to stimulation of CD36 and GPR120 [38]. In addition, the expression of CD36 is regulated via PPAR gamma and alpha, which are both activated by endocannabinoids [73,74,75]. Inversely, CD36 signalling regulates the release of peripheral endocannabinoids, and OEA has anti-orexigenic features [76,77,78] and stimulates lipolysis [79]. We did not observe group differences in endocannabinoids in plasma or the tongues suggesting that endocannabinoids do not explain the fat preference of Grn^−/−^ mice, yet the signalling of satiety by OEA might be lower in mice with low expressions of CD36 [77].

We conclude that progranulin deficiency resulted in the loss of fat sense and, therefore, the high attraction of mice to fat via the downregulation of the receptor, CD36. The inverse translation suggests that progranulin increases the gustatory sense of fat, thereby producing anti-orexigenic effects, in line with a previous study [80]. The result appears to contradict observations of high progranulin serum levels in obesity [20,21], which do not reveal whether progranulin is a driver of fat intake or an attempt to stop overeating and reduce obesity-associated inflammation. Since we used a global progranulin knockout, our results did not reveal if the downregulation of CD36 in taste buds was a local or systemic effect. It is of note that similar controversial results were reported for progranulin’s associations with cardiovascular disease. Serum levels were positively associated with microvascular inflammation [81] but knockouts developed stronger cardiovascular diseases with apolipoprotein E deficiency [26], and progranulin expression was low in cardiovascular lesions [82].

## 5. Conclusions

In summary, we showed that progranulin deficiency urged mice to take up fat nutrients due to the association with low levels of CD36 in taste buds, likely increasing the urge for fatty foods. When applied to humans, the data suggest that patients suffering from diseases associated with low progranulin levels might have a high preference for fat taste. Compulsive eating is one of the behavioral abnormalities in Frontotemporal Dementia, a disease caused by loss-of-functions progranulin mutations. However, low progranulin plasma levels were also observed in patients with psychiatric diseases and a high comorbidity of obesity [83,84]. One may speculate that CD36-ligand lolly might possibly help to reduce the appeal of the high-fat taste.

## Figures and Tables

**Figure 1 nutrients-13-04125-f001:**
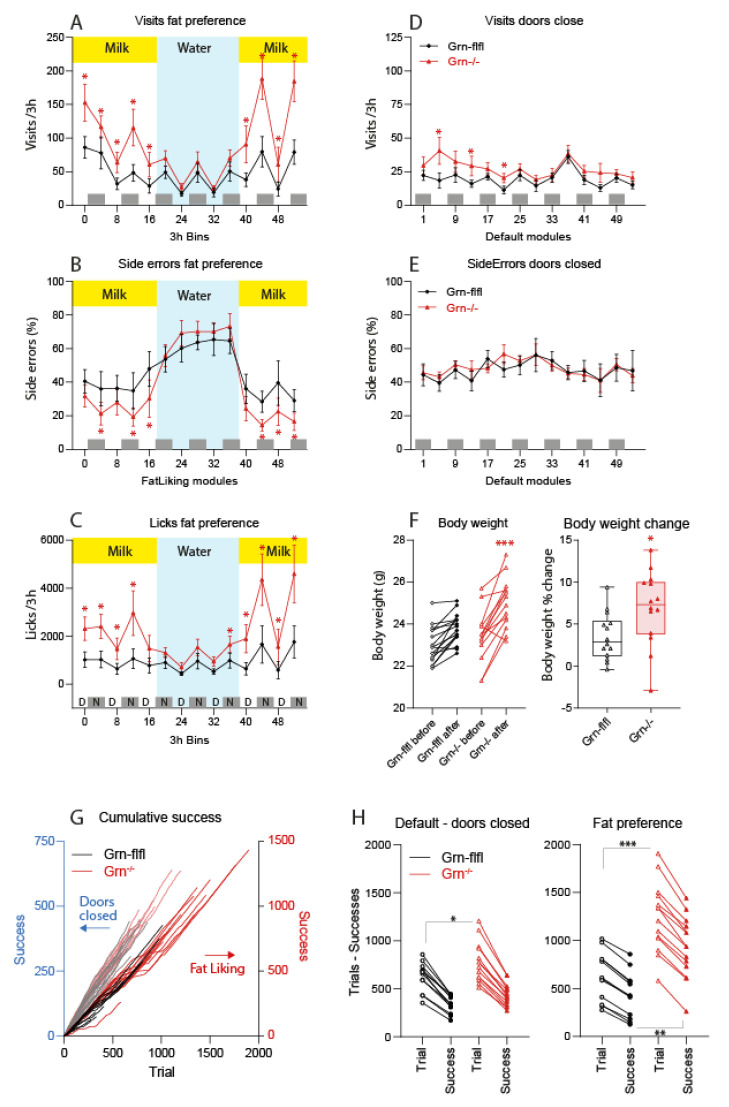
IntelliCage behavior of progranulin-deficient and control mice in a fat preference test. Female progranulin knockout (Grn^−/−^, triangle red) and control mice (Grn-flfl, circle black) were housed in two IntelliCages, in each cage *n* = 7–8 per genotype, age 28–37 weeks at the start of the experiment. In the fat taste preference module, mice could freely choose between 0.3% skimmed milk on one side and 2% milk on the other in each corner. This module was active 2 × 3 h each day (11:00–14:00 and 2:00–5:00 a’clock). In between, doors remained closed (default module). After 3 days, the bottles were switched to tap water for 2 days, and back to milk for another 2 days. Side errors are defined as nosepoke on the skimmed milk side. (**A**–**C**): Time courses of visits, side errors and licks during in 3 h intervals in the active fat taste preference module. The data show the mean ± 95% CI. The grey patterned bar in the bottom indicates daytime and night time behavior. (**D**,**E**): Time course of visits and side errors in the default module where doors remained closed, and licks were not allowed. Time course data were compared with two-way ANOVA for “time” X “genotype” and subsequent posthoc analysis using an adjustment of alpha according to Šidák. (**F**): Body weights before and after the IntelliCage experiment and net change of the body weight. The boxes show the interquartile range, the line is the median, whiskers are minimum to maximum, the scatters show the mice. Data were compared per two-way ANOVA (paired data plot) and unpaired, two-tailed Student’s *t*-test. (**G**): Trials (visit) versus successes (correct nosepoke on the 2% milk side) in the fat taste preference (fat liking, right *Y*-axis) and the default module (left *Y*-axis) in which the doors remained closed. Grn^−/−^ mice made more trials and had more successes. (**H**): Paired data plots of trials and successes in the default module (doors closed) and fat taste preference module (doors opened on NP). Data were compared with two-way ANOVA for “Module” X “genotype” and subsequent Šidák posthoc analysis. Asterisks indicated statistically significant differences between genotypes. * *p* < 0.05, ** *p* < 0.01, *** *p* < 0.001.

**Figure 2 nutrients-13-04125-f002:**
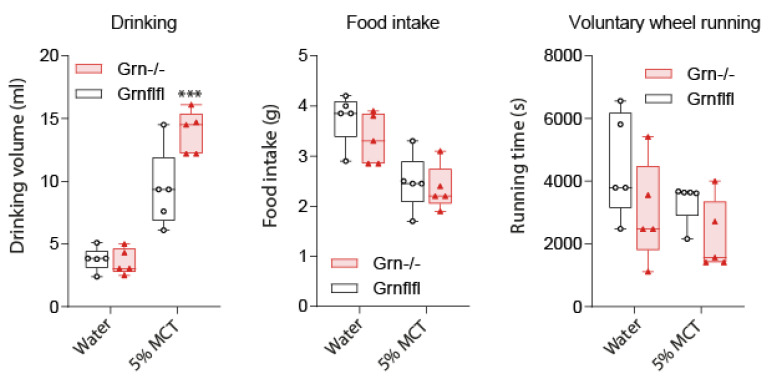
Drinking, feeding and voluntary wheel running with tap water or MCT water. Female progranulin knockout (Grn^−/−^, triangle red) and control mice (Grn-flfl, circle black) were housed in two Phenomaster cages in pairs of two mice (*n* = 5 per genotype, 9–10 weeks old). They had free access to the drinking bottle, feeding basket and voluntary running wheel. In the first 24 h period, the bottle was filled with tap water, in the second with a 5% medium chain triglyceride (MCT) solution in water. The boxes show the interquartile range, the line is the median, whiskers are minimum to maximum, the scatters show the mice. Data were compared with unpaired, two-tailed Student’s *t*-tests. *** *p* < 0.001.

**Figure 3 nutrients-13-04125-f003:**
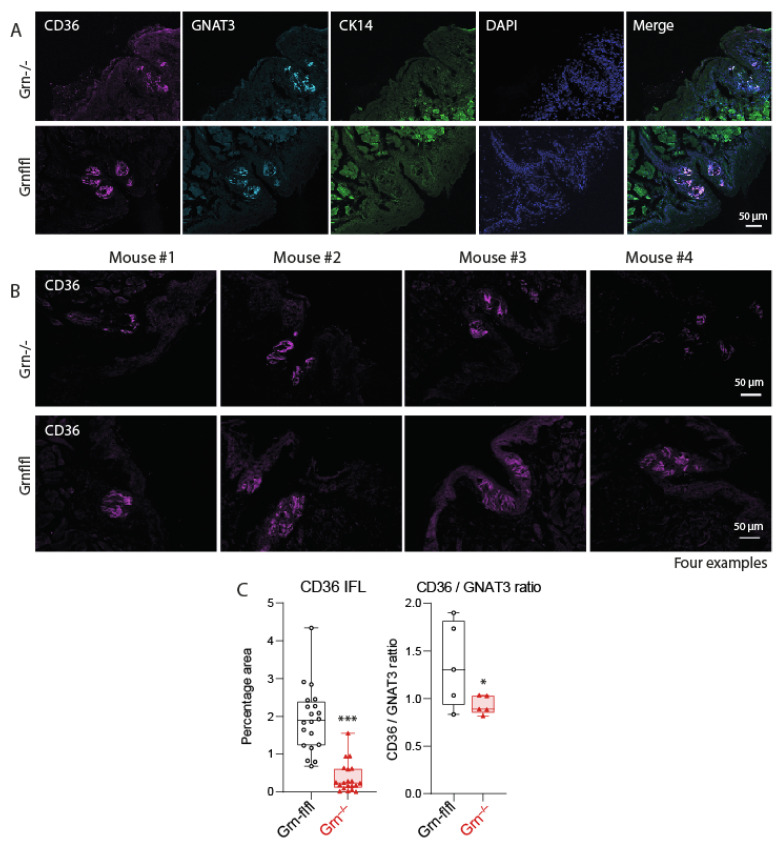
Immunofluorescence analysis and QRT-PCR analysis of CD36 in taste buds. (**A**): Immunofluorescence analysis of CD36, the taste bud markers alpha-Gustducin (GNAT3) and cytokeratin-14 (CK14) in circumvallate papillae of Grn^−/−^ (triangle red) and Grn-flfl (circle black) mice. DAPI was used as nuclear counterstain. Scale bars are 50 µm. (**B**): Examples of CD36 immunofluorescence in circumvallate papillae of each four Grn^−/−^ and Grn-flfl mice. Further mice in Appendix A. Scale bars are 50 µm. (**C**): Quantification of CD36 immunofluorescence expressed as percentage area of CD36 immunoreactivity (*n* = 8–9 per groups) and as CD36/GNAT3 ratio (*n* = 5 per group). The area was obtained with the Particle Counter in FIJI ImageJ. The box is the interquartile range, the line is the median, the whiskers show minimum to maximum, the scatter are images of *n* = 8–9 mice per group (further images in Appendix A). Data were compared with unpaired, two-tailed Student’s *t*-tests. * *p* < 0.05, *** *p* < 0.001.

**Figure 4 nutrients-13-04125-f004:**
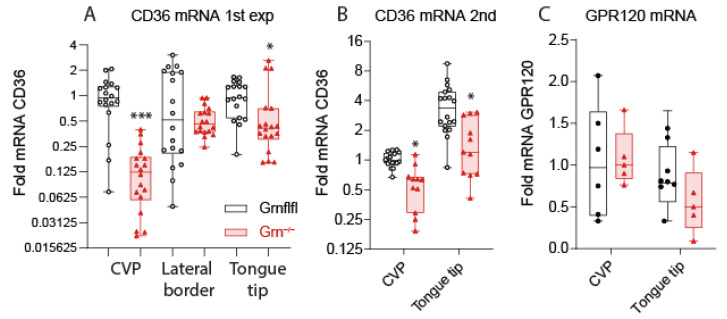
Quantitative RT-PCR analysis of CD36 and GPR120 in mouse tongue. (**A**,**B**): QRT-PCR analysis of CD36 in the tongue of Grn^−/−^ (trinagle red) and Grn-flfl (circle black) mice of the CVP region, lateral border and tongue tip in two independent experiments (**A**,**B**), each comprising of n = 6 mice per genotype and triplicate analyses per mouse (total sample size per genotype *n* = 12 mice). EEF1G was used as housekeeping gene for normalization of the cycle numbers. Data were analyzed according to the ∆∆Ct method (**C**): QRT-PCR analysis of GPR120 in the tongue of Grn^−/−^ and Grn-flfl of the CVP region and the tongue tip. The expression of GPR120 was low and some samples had cycle numbers above the upper threshold of 40 cycles. The presented data are of *n* = 6 mice per genotype and triplicate analyses. Data show the fold difference versus the CVP mean of Grn-flfl control mice and were compared with 2-way ANOVA for “region” × “genotype” and subsequent posthoc analysis for genotype according to Šidák. * *p* < 0.05; *** *p* < 0.001. Endocannabinoids, ceramides and sphingolipids in tongue tissue.

**Figure 5 nutrients-13-04125-f005:**
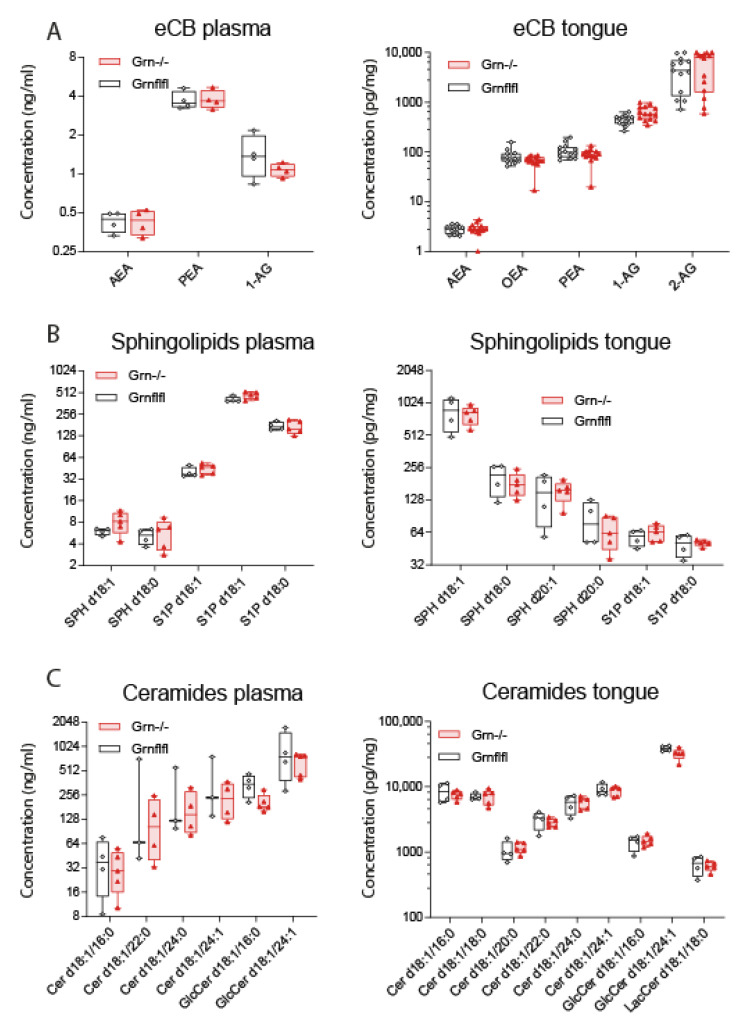
Endocannabinoids, sphingolipids and ceramides in plasma and tongue. (**A**): Endocannabinoids (eCB) in plasma and tongue in Grn^−/−^ (triangle red) and Grn-flfl (circle black) mice (11–12 weeks, *n* = 5 for plasma and *n* = 14 for tongue). AEA, anandamide; OEA, oleoylethanolamide; PEA, palmitoylethanolamide; 1-AG/2-AG, arachidonoylglycerol. (**B**): Sphingolipids of different chain length, hydroxylation and phosphorylation in plasma and tongue in Grn^−/−^ and Grn-flfl mice. SPH d18:0, sphinganine; SPH d18:1 sphingosine; S1P d18:0 sphinganine-1-phosphate; S1P d18:1 sphingosine-1-phosphate. (**C**): Ceramides and hexosylceramides of different chain length and saturation in plasma and tongue in Grn^−/−^ and Grn-flfl mice. Cer d18:1/16:0, 16C ceramide without saturation; GlcCer, glucosylceramide; LacCer, lactosylceramide. The box is the interquartile range, the line is the median, the whiskers show minimum to maximum, the scatters are mice. Data were compared with two-way ANOVA for “lipid” × “genotype” for each lipid class. Sphingolipid and ceramide patterns were not affected by genotype.

## Data Availability

Data are presented in the paper. Raw data shall be made available on reasonable request. No omics datasets were generated or analyzed during the current study.

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
