# Peer review of "Increased Fat Taste Preference in Progranulin-Deficient Mice"

_nutrients, 2021, doi:10.3390/nu13114125_

Round 1
Reviewer 1 Report
In this manuscript, the authors demonstrate that the increased fat taste preference in progranulin deficient mice by the down regulated CD36 expression. I also think that the topic may be of interest to the field. The data are novel and informative. Overall, the manuscript is interesting, and the data support the main conclusions. However, I think that the immunofluorescence analysis is problematic.
In Fig. 3A, the expression levels of CD36 per GNAT3 expression looks like same between Grn-/- mice and Grnflfl mice. It is important to analyze the expression percentages of CD36 in GNAT3 positive cells between two mouse lines. GNAT3 is a marker of type 2 taste cell.
In Fig. 3B and Suppl. Fig S2, there are no internal controls such as GNAT3. So, we can’t judge the decrease of CD36 expression. It is important to normalize the CD36 expression by using other taste cell marker. If the authors have not data about these taste cell markers, it is better to write the limitation in discussion. But, because there are qRT-PCR data in Fig. 3D, I think the conclusion of this manuscript is correct.
Author Response
Reviewer #1
In this manuscript, the authors demonstrate that the increased fat taste preference in progranulin deficient mice by the down regulated CD36 expression. I also think that the topic may be of interest to the field. The data are novel and informative. Overall, the manuscript is interesting, and the data support the main conclusions. However, I think that the immunofluorescence analysis is problematic.
Thank you for evaluation of our manuscript and providing us with your insight.
The immunofluorescence of CD36 in the taste buds is challenging and the images are convincing with high sample sizes (all presented) and they clearly reveal the structure of the taste buds. CD36 immunoreactivity specifically occurred in the taste buds. The images clearly show that there was no unspecific signal of surrounding tissue. We do not agree that the analysis of CD36/GNAT3 signals was/is mandatory. Indeed, quantification of immunofluorescence studies is often more reliable in mono-stainings that avoid antibody interferences and fluorescence bleed through. The images are of high quality and we do not agree that the immunofluorescence analysis is "problematic".
In Fig. 3A, the expression levels of CD36 per GNAT3 expression looks like same between Grn-/- mice and Grnflfl mice. It is important to analyze the expression percentages of CD36 in GNAT3 positive cells between two mouse lines. GNAT3 is a marker of type 2 taste cell.
GNAT3 was stronger or equal in Grn-/- mice as compared to control mice so that the CD36/GNAT3 ratio is lower than that of the control mice. We have added the quantification of the CD36/GNAT3 ratio in Figure 3C of those mice (n = 5 per group) from which we had obtained these double stainings.
In Fig. 3B and Suppl. Fig S2, there are no internal controls such as GNAT3. So, we can’t judge the decrease of CD36 expression. It is important to normalize the CD36 expression by using other taste cell marker. If the authors have not data about these taste cell markers, it is better to write the limitation in discussion. But, because there are qRT-PCR data in Fig. 3D, I think the conclusion of this manuscript is correct.
Please see the response to your comment above. Text edits are highlighted by WORD's track changes. Figure 3 is replaced with a new Figure 3 including CD36/GNAT3 ratios.
Reviewer 2 Report
General comments
The paper “Increased fat taste preference in progranulin deficient mice” provides evidence of the effects of progranulin deficiency on feeding habits on fat nutrients in rats, thus suggesting that diseases associated with low progranulin levels might have a high preference for fat taste. The results presented are robust and novel, although they are a bit contradictory with those of other papers, as recognized by the Authors themselves in the Discussion.
The are some problems regarding the “organization” of the paper. On the basis of the concepts expressed, some sentences are not properly located and should be moved to different sections (see below).
Introduction
Lines 68-70. In this context, I suggest to add a reference also to these papers:
Sollai et al. (2019). Human tongue electrophysiological response to oleic acid and its associations with PROP taster status and the CD36 polymorphism (rs1761667). Nutrients, 11(2), 315. DOI: 10.3390/nu11020315.
Melis et al. (2015). Associations between Orosensory Perception of Oleic Acid, the Common Single Nucleotide Polymorphisms (rs1761667 and rs1527483) in the CD36 Gene, and 6-n-Propylthiouracil (PROP) Tasting. Nutrients, 7: 2068-2084. doi:10.3390/nu7032068.
Materials and Methods
Line 116. The sentence “The nosepoke-protocol was effective from 11-14:00 h and 2-5 h each day” is not clear. Do the Authors mean: “The nosepoke-protocol started at 11:00–14:00 h each day and lasted 2-5 hours”?
Lines 124-125. In the measurement of body weight, have the authors taken into account differences in the physical activity of mice? Besides, have they measured the amount of milk, water and diet pellets actually ingested? This information should be provided in the Materials and Methods Section.
Line 127. “Each five 9-10 weeks old…”. What does this sentence mean?
Line 134. MCT = Medium-Chain (instead of middle-chain) Triglycerides.
Lines 135-136. Have water evaporation phenomena been taken into account?
Line 158. It is not clear whether the mice in each experimental set belong to different groups or not. If the latter hypothesis is correct, this could be a limiting factor for the study: different animals could affect the results, even if standardized. Furthermore, it is not clear why the behavioural tests were performed only on females (Line 113), while the genetic and immunohistochemical analyses also on males.
Results
Lines 229-234. Trovo che questa parte sia una descrizione della procedura sperimentale, per cui andrebbe spostata nei M&M dove, per di più, la descrizione lascia molto a desiderare. This sentence should be part of the description of the experimental procedure, so it should be moved to the Materials and Methods. Besides the description presents the same understanding problems as the section at lines 118-123.Lines 246-248. I find this sentence more appropriate in the Discussion than in the Results.
Line 257. What is 28-37: days or weeks?
Lines 260-262. This has been written already in the Materials and Methods.
Lines 263-280. This paragraph is formatted as the caption of Figure 1. It is not the proper way of describing the results. I believe the authors should re-write this section to make it more readable.
Line 283. MCT = Medium-Chain (instead of mid-chain) Triglycerides.
Lines 293-299. This sentence is not appropriate in the Results: it should be in the Materials and Methods.
Lines 301-304. The concept expressed in the introduction of this sentence would be more appropriate in the Introduction.
Lines 316-328. Again: this text is structured more as a caption to Figure 3 than a part of the Results.
Lines 330-334. This sentence is more a conclusion than a result and should be moved to the Discussion.
Lines 343-355. As above (Lines 316-328), this sentence reads more like a caption to Figure 4 than the presentation of results.
Discussion
Line 403. The meaning of the acronym APOE (Apolipoprotein E) is nowhere explained in the text.
Lines 408-409. This sentence is not clear! What do you mean by saying that diseases might have a high preference? Please reword.
Author Response
Reviewer # 2
The paper “Increased fat taste preference in progranulin deficient mice” provides evidence of the effects of progranulin deficiency on feeding habits on fat nutrients in rats, thus suggesting that diseases associated with low progranulin levels might have a high preference for fat taste. The results presented are robust and novel, although they are a bit contradictory with those of other papers, as recognized by the Authors themselves in the Discussion.
The are some problems regarding the “organization” of the paper. On the basis of the concepts expressed, some sentences are not properly located and should be moved to different sections (see below).
Thank you for evaluation of our manuscript and providing us with your suggestions. We have edited the text as suggested. Please see the specific responses below. The text edits are highlighted by WORD's track changes.
Introduction
Lines 68-70. In this context, I suggest to add a reference also to these papers:
Sollai et al. (2019). Human tongue electrophysiological response to oleic acid and its associations with PROP taster status and the CD36 polymorphism (rs1761667). Nutrients, 11(2), 315. DOI: 10.3390/nu11020315.
Melis et al. (2015). Associations between Orosensory Perception of Oleic Acid, the Common Single Nucleotide Polymorphisms (rs1761667 and rs1527483) in the CD36 Gene, and 6-n-Propylthiouracil (PROP) Tasting. Nutrients, 7: 2068-2084. doi:10.3390/nu7032068.
We have added these references as requested.
Materials and Methods
Line 116. The sentence “The nosepoke-protocol was effective from 11-14:00 h and 2-5 h each day” is not clear. Do the Authors mean: “The nosepoke-protocol started at 11:00–14:00 h each day and lasted 2-5 hours”?
The nosepoke protocol was ON from 11:00 to 14:00, and then again ON from 2:00 to 5:00. In between the protocol was OFF and a default protocol was running in which doors remained closed (i.e. did not open upon nosepoking at the doors). It is reworded.
Lines 124-125. In the measurement of body weight, have the authors taken into account differences in the physical activity of mice? Besides, have they measured the amount of milk, water and diet pellets actually ingested? This information should be provided in the Materials and Methods Section.
Physical activity was monitored by analysis of the frequency of the visits in the IntelliCages in the periods when the doors were closed (Default modules: Figure 1 D and E). There was no difference. Activity was also monitored in the Phenomaster Cages which have a Voluntary Running Wheel (Figure 2). There was no significant difference. Food ingestion was assessed in the Phenomaster Cages (Figure 2). Again, there was no difference between genotypes. In the IntelliCages 16 mice are living together in one cage and it is not possible to monitor the individuals' food intake. For drinking, the licking numbers tell about the drinking volume, which is however not measured as a volume but as number of lickings. The lickings are presented in Figure 1C and in Suppl. Fig. 1, second row.
Line 127. “Each five 9-10 weeks old…”. What does this sentence mean?
Five mice were used per group, and they were 9-10 weeks old. The sentence is now reworded.
Line 134. MCT = Medium-Chain (instead of middle-chain) Triglycerides.
We have corrected this error.
Lines 135-136. Have water evaporation phenomena been taken into account?
The bottles are tightly closed with a narrow nipple. Evaporation does not occur.
Line 158. It is not clear whether the mice in each experimental set belong to different groups or not. If the latter hypothesis is correct, this could be a limiting factor for the study: different animals could affect the results, even if standardized. Furthermore, it is not clear why the behavioural tests were performed only on females (Line 113), while the genetic and immunohistochemical analyses also on males.
The mice were progranulin knockout mice versus age and gender matched controls in each experiment. Tissue was obtained from all mice who participated in the behavioural studies. But it is not possible to do all biological studies with the behavioural cohorts because the tissue for immunofluorescence and rt-PCR has to be prepared differently. I do not understand the issue with Line 158.
In the IntelliCages 16 mice are living together, and males are fighting which impairs IntelliCage observations. Therefore, female mice were used which is standard procedure for IC observations.
Results
Lines 229-234. Trovo che questa parte sia una descrizione della procedura sperimentale, per cui andrebbe spostata nei M&M dove, per di più, la descrizione lascia molto a desiderare.
Sorry, but we do not speak Italian. It appears to be a left-over of a second person reading the manuscript.
This sentence should be part of the description of the experimental procedure, so it should be moved to the Materials and Methods. Besides the description presents the same understanding problems as the section at lines 118-123.Lines 246-248. I find this sentence more appropriate in the Discussion than in the Results.
We have moved some sentences as suggested, but think that a short repetition of the behavioural set up helps to follow the Results.
Line 257. What is 28-37: days or weeks?
"Weeks" was added.
Lines 260-262. This has been written already in the Materials and Methods.
The text is now reduced in the Results
Lines 263-280. This paragraph is formatted as the caption of Figure 1. It is not the proper way of describing the results. I believe the authors should re-write this section to make it more readable.
This is the figure caption/figure legend of Figure 1, and it is formatted according to the journal's style as figure legend.
Line 283. MCT = Medium-Chain (instead of mid-chain) Triglycerides.
Corrected, please see above
Lines 293-299. This sentence is not appropriate in the Results: it should be in the Materials and Methods.
This is the Figure Legend of Figure 2 and it is correctly formatted as a Figure legend according to the Journal's style.
Lines 301-304. The concept expressed in the introduction of this sentence would be more appropriate in the Introduction.
The concept is introduced in the last sentence of the Introduction. It is briefly repeated as intro to the CD36 results, which is normally considered as good scientific writing style so that the reader can easily follow the rationale for the next experiments.
Lines 316-328. Again: this text is structured more as a caption to Figure 3 than a part of the Results.
Again, this is the figure legend and it is formatted according to journal style.
Lines 330-334. This sentence is more a conclusion than a result and should be moved to the Discussion.
We find this sentence well placed as concluding remark for the results in this section.
Lines 343-355. As above (Lines 316-328), this sentence reads more like a caption to Figure 4 than the presentation of results.
Again, this is the figure legend formatted according to the Journal's style
Discussion
Line 403. The meaning of the acronym APOE (Apolipoprotein E) is nowhere explained in the text.
It is now explained.
Lines 408-409. This sentence is not clear! What do you mean by saying that diseases might have a high preference? Please reword.
Translated to human, the data suggest that patients suffering from diseases associated with low progranulin levels might have a high preference of fat taste.